# Inhaled Argon Impedes Hepatic Regeneration after Ischemia/Reperfusion Injury in Rats

**DOI:** 10.3390/ijms21155457

**Published:** 2020-07-30

**Authors:** Sophia M. Schmitz, Henriette Dohmeier, Christian Stoppe, Patrick H. Alizai, Sandra Schipper, Ulf P. Neumann, Mark Coburn, Tom F. Ulmer

**Affiliations:** 1Department of General, Visceral and Transplantation Surgery, University Hospital of RWTH Aachen, Pauwelsstr. 30, 52074 Aachen, Germany; sopschmitz@ukaachen.de (S.M.S.); palizai@ukaachen.de (P.H.A.); sschipper@ukaachen.de (S.S.); uneumann@ukaachen.de (U.P.N.); 2Department of Anaesthesiology, University Hospital of RWTH Aachen, Pauwelsstr. 30, 52074 Aachen, Germany; hdohmeier@ukaachen.de (H.D.); cstoppe@ukaachen.de (C.S.); mcoburn@ukaachen.de (M.C.); 3Department of Nanomedicine and Theranostics, Institute for Experimental Molecular Imaging, University Hospital of RWTH Aachen, Faculty of Medicine, RWTH Aachen University, 52074 Aachen, Germany; 4Department of Surgery, Maastricht University Medical Centre (MUMC), P.O. Box 5800, 6202 AZ Maastricht, The Netherlands

**Keywords:** liver surgery, transplantation, ischemia/reperfusion, injury, liver regeneration, argon, interleukin 6

## Abstract

Organoprotective effects of noble gases are subject of current research. One important field of interest is the effect of noble gases on hepatic regenerative capacity. For the noble gas argon, promising studies demonstrated remarkable experimental effects in neuronal and renal cells. The aim of this study was to investigate the effects of argon on the regenerative capacity of the liver after ischemia/reperfusion injury (IRI). Male, Sprague-Dawley rats underwent hepatic IRI by clamping of the hepatic artery. Expression of hepatoproliferative genes (HGF, IL-1β, IL-6, TNF), cell cycle markers (BrdU, TUNEL, Ki-67), and liver enzymes (ALT, AST, Bilirubin, LDH) were assessed 3, 36, and 96 h after IRI. Expression of IL-1β and IL-6 was significantly higher after argon inhalation after 36 h (IL-1β 5.0 vs. 8.7 fold, *p* = 0.001; IL-6 9.6 vs. 19.1 fold, *p* = 0.05). Ki-67 was higher in the control group compared to the argon group after 36 h (214.0 vs. 38.7 positive cells/1000 hepatocytes, *p* = 0.045). Serum levels of AST and ALT did not differ significantly between groups. Our data indicate that argon inhalation has detrimental effects on liver regeneration after IRI as measured by elevated levels of the proinflammatory cytokines IL-1β and IL-6 after 36 h. In line with these results, Ki-67 is decreased in the argon group, indicating a negative effect on liver regeneration in argon inhalation.

## 1. Introduction

Ischemia-reperfusion injury (IRI) is an undesirable consequence of organ transplantation and frequently occurs in other clinical settings such as major liver surgery or circulatory shock after resuscitation [1,2,3]. It is one of the most important determinants of organ dysfunction (i.e., liver failure) and determines short- and long-term morbidity and mortality after liver resection and transplantation [4,5,6].

Extensive research has been undertaken to discover and establish strategies to reduce or prevent hepatic IRI [7,8]. Recently, the use of noble gases has gained attention. Previous studies showed some remarkable cytoprotective effects of noble gases in different clinical settings. For instance, helium was associated with cytoprotective effects in kidney cells and after traumatic brain injury [9,10]. For the noble gas xenon, neuroprotective effects have also been described [11,12], while anti-inflammatory effects have been demonstrated in cardiac surgery patients [13] Concerning the effect of these noble gases on the liver, helium does not appear to have a protective effect in hepatic IRI [14]. In contrast, some studies suggest a slightly positive effect of xenon on the liver function [15,16]. The noble gas argon seems to have similar effects at least with regard to IRI in kidney transplantation [17]. Furthermore, Argon was reported to be associated with cytoprotective effects after neuronal, retinal, and cardiac injury following ischemic stress [10,18,19,20,21,22,23,24]. Of interest is that the noble gas argon is cheaper and wider available in comparison to xenon or helium. All of these characteristics make the noble gas argon a promising candidate for reducing hepatic IRI. Nevertheless, the influence of argon on hepatic IRI has not been investigated yet. Therefore, the goal of the present study is to evaluate the effect of argon on IRI in an animal model of hepatic injury. 

## 2. Results

### 2.1. Effect of Argon on Hepatocytes Proliferation and Apoptosis after I/R 

The amount of Ki67 positive cells per 1000 hepatocytes in the control group was significantly increased compared to all other conditions at 36 h (F (7,50) = 3.922, *p* = 0.0018; ARG 36 h vs. CO 36 h: *p* = 0.045).

The same pattern was observed for BrdU except that there was no difference between the group receiving argon and the control group (F (7,49) = 4.162, *p* = 0.001, ARG 36 h vs. CO 36 h *p* = 0.455, significance bars are not indicated in Figure 1).

There were no significant differences for the number of TUNEL positive cells between the different groups (F (7,60) = 1,891, *p* = 0.0869). 

Stainings for BrdU, Ki67, and TUNEL are shown in Figure 2.

### 2.2. Argon Modulates the Expression of Cytokines 

There were significant differences in the concentration of HGF (F (7,46) = 10.17, *p* < 0.0001), IL-1β (F (7,45) = 39.29, *p* < 0.001), IL-6 (F (7,46) = 17.26, *p* < 0.001) and TNF-α (F (7,44) = 33.36, *p* < 0.001) between the different time points and groups. Between the control and argon group, there was a significant difference at 36 h for IL-1β (*p* = 0.001) and IL-6 (*p* = 0.050) (see Figure 3) at 36 h. There were no differences between argon and control group for HGF and TNF-α at any time point. 

### 2.3. Plasma Biochemical Data and Parameters of Liver Function (Liver Specific Values)

One-way ANOVA analysis revealed a significant difference for the following parameters between the various time points: AP (F (7,51) = 9.653, *p* < 0.0001), AST (F (7,49) = 10.51, *p* < 0.0001), ALT (F (7,50) = 10.29, *p* < 0.0001), LDH (F(7,48) = 8.610, *p* < 0.0001), and protein content (F (7,51) = 3.660, *p* = 0.0028). Yet, none of these values as well as Bilirubin and total protein count differed significantly between the argon and control group (see Figure 4).

### 2.4. Liver Injury after IRI Measured by Suzuki’s Score

HE stainings and Suzuki’s score revealed significant liver injury starting 3h after IRI with a peak after 36 h (see Figure 5 and Figure 6). There was no difference between argon and control group at any time point. 

## 3. Methods

The protocol for this study was approved by the government agency for animal use and protection (Protocol number: 84-02.04.2012.A014 approved by “Landesamt für Natur, Umwelt und Verbraucherschutz NRW” (LANUV), Recklinghausen, Germany). All experiments were performed in accordance with the Guide for the Care and Use of Laboratory Animals (National Research Council (US), and the Committee for the Update of the Guide for the Care and Use of Laboratory Animals; 8th edition, 2011).

### 3.1. Animals and Study Design

Experiments were carried out with male, Sprague-Dawley (SD) rats (Charles River, Sulzfeld, Germany), aged 12–14 weeks old. The animals were housed for at least one week before surgery with access to standard diet and water ad libitum on a 12-h light/dark cycle under SPF (specific pathogen free) conditions. Animals were randomly assigned to treatment groups either undergoing sham surgery with argon inhalation (sham experimental group, ARG Sham, *n* = 8) or without argon inhalation (CO Sham, *n* = 8), ischemia/reperfusion (I/R) injury with argon inhalation (ARG3, ARG36, ARG96, *n* = 8 per group and time point) or I/R without argon inhalation (CO3, CO36, CO96, *n* = 8 per group and time point). Rats were sacrificed after 3, 36 or 96 h after the operation. Survival times were determined according to a previously performed experiment [25].

### 3.2. Anesthesia 

All surgical procedures were performed under inhalation anesthesia as previously described [26]. Prior to the operation, animals were exposed to either 50 Vol% argon (argon group) or 50 Vol% nitrogen (control group) balanced with 48 Vol% oxygen and 2 Vol% Isoflurane for 60 min (preconditioning). Concentrations were based on the findings of a previous study [25]. Directly thereafter, surgery was performed under isoflurane anesthesia (1.5 Vol%) and the argon inhalation was stopped. Gas mixtures were administered with a total flow of 700 mL/min and animals were breathing spontaneously through a rat facemask (ZUA-82-ANÄ-MAS-MA/RA-MMH; Föhr Medical Instruments GmbH), which covered the mouth and snout of the animals. Argon and nitrogen were provided with a purity ≥99.998% (Linde AG, Pullach, Germany). A heating pad maintained the body temperature of the animals at 37–37.5 °C (Fine Science Tools Inc, Heidelberg, Germany).

### 3.3. Surgical Procedure and Postoperative Care

Under sterile conditions, hepatic IRI was performed. Following a median laparotomy, an atraumatic clip (Aesculap, Tuttlingen, Germany) was applied with a clip applicator to interrupt arterial and portal venous blood supply to the left and middle lobes (70% liver volume). Discoloration of the liver indicated correct positioning of the clip. Using this method of partial hepatic ischemia, portal vein congestion and subsequent bacterial translocation into the portal venous blood was avoided. After 45 min, clamps were removed, and reperfusion was started. Once the liver regained color the abdominal cavity was closed using continuous sutures in double-layer technique. In sham-operated controls, all procedures were identical, but no vascular occlusion was performed. Postoperatively, gas delivery was discontinued, the facemask removed, and animals were breathing ambient air spontaneously and were returned to their heated cages. Once they were mobile, rats received postoperative analgesia (Buprenorphine 0.03 mg/kg body weight) subcutaneously. Animals were sacrificed 3, 36 and 96 h after IRI under terminal anesthesia (isoflurane) by exsanguination. Two hours before sacrifice, 200 μL BrdU (3 mg/mL) was injected intrapertioneally. 

### 3.4. Histology

The whole liver was removed and half of it methanol-stabilized formaldehyde (3.5%) was used for emersion fixation. The other part was spared for RNA analysis. Next, the tissue was dehydrated and embedded in paraffin and cut in 3-μm sections on a microtome and mounted on slides. Paraffin-embedded liver sections for immunohistochemistry were immersed in three changes of xylene and hydrated with a graded series of alcohol. Hematoxylin and eosin staining (H&E) was used to evaluate the regenerative capacity of the liver by determination of Mitotic Index and the liver damage by Suzuki’s score [27]. Suzuki’s score was obtained by a single blinded pathologist experienced in liver histology. 

### 3.5. Immunohistochemistry

Immunohistochemistry was performed on paraffin-embedded liver sections. Microwave treatment was used for unmasking antigens. Pathologic quantification was conducted independently by two experienced pathologists, blinded to the treatment group. In each specimen, positive cells were counted in five randomly selected fields (40x). 

#### 3.5.1. Bromodeoxyuridine 

For determining the number of hepatocytes undergoing DNA synthesis, bromodeoxyuridine (BrdU) was used. Sections were incubated with a monoclonal mouse anti-BrdU antibody (1:200; Dako, Clone Bu20a, Glostrup, Denmark) over night at 4 °C. Consequential, sections were incubated with secondary anti-mouse-antibody (1:300, Dako, Glostrup, Denmark) and sections were treated with an avidin-biotin-peroxidase system (ABC kit; Vector Laboratories, Inc., Burlingame, CA). 3,3′diaminobenzidine (DAB) (Sigma-Aldrich, Steinheim, Germany) was used for detecting the sites of peroxidase-binding and the sections were counterstained with Haematoxylin according to standard protocol. In total, 1000 nuclei of hepatocytes were counted (per field of view) and the fraction of BrdU positive cells was determined. 

#### 3.5.2. KI-67

The cell cycle marker Ki-67 was analyzed using a combination of primary monoclonal mouse antibody (1:10; Dako; Glostrup, Denmark) and secondary biotinylated rabbit anti-mouse antibody (1:300; Dako; Glostrup, Denmark). For visualization, DAB (Sigma-Aldrich; Steinheim, Germany) was used and nuclei were counterstained using haematoxylin (Sigma-Aldrich; Steinheim, Germany).

#### 3.5.3. TUNEL

Terminal deoxynucleotidyl transferase dUTP nick end labeling (TUNEL) was used to quantify the number of apoptotic cells by using the in situ apoptosis detection kit (ApopTag Peroxidase Kit, S7100; Intergen, Oxford, UK). All steps were performed according to the supplier’s instruction. Labeled DNA fragments were visualized by DAB (Sigma-Aldrich Steinheim, Germany). Nuclei were stained with haematoxylin (Sigma-Aldrich, Steinheim, Germany).

### 3.6. RNA Isolation and Quantitative RT-PCR 

Gene expression of pro- and anti-inflammatory cytokines Interleukin-1β (IL-1β), Interleukin-6 (IL-6) and Tumor Necrosis Factor-α (TNF-α) at the specified time points was measured as follows: RNA was extracted with the Nucleospin RNA-II Kit (Macherey-Nagel, Dueren, Germany) from cryopreserved, homogenized liver tissue. Afterwards conversion into cDNA was performed by using the Omniscript reverse transcriptase kit (Qiagen, Hilden, Germany). A total of 1500 ng RNA was used. All procedures were performed in accordance with manufacturers guidelines. 

The gene expression analysis was performed on an ABI 7500 Real-Time PCR system (Life Technologies, Darmstadt, Germany). Quantitative expression of HGF, IL-1β, IL-6 and TNF-α was analyzed. Murine qPCR Primers were designed using the Primer BLAST tool provided by the NCBI website. The applied primer pairs were exon-exon junction spanning or at least separated by an intron. The design strategy included secondary structure analysis (mfold analysis), a species-wide specificity check, and an evaluation of potential homo- and heterodimerization of primers. SYBR-Green technology (Applied Biosystems, California, USA) was used for detecting the PCR products. Melt curve analysis and TAE-buffered DNA agarose gel electrophoresis of the PCR product were used to determine primer specificity. Amplification efficiency was calculated using LinRegPCR 2015.3 (Heart Failure Research Center, Amsterdam, The Netherlands) as previously described [28]. Target gene expression was normalized by a reference gene index including the expression of β2-microglobulin (B2M), β-actin (ActB) and ribosomal protein L13a (RPL13A). These genes were determined as the most suitable for normalization in this study. This reference gene evaluation using geNorm calculation (as part of the qbase+ software) was performed beforehand. Qbase+ 2.6.1 (Biogazelle, Zwijnaarde, Belgium) was used for calculating changes in gene expression [29].

### 3.7. Blood Samples 

After sacrifice approximately 0.6 mL of blood was collected from the heart of the rats. The blood was centrifuged and serum was stored at –70.8 °C in a refrigerator. Serum levels of aspartate aminotransferase (AST), alanine aminotransferase (ALT), alkaline phosphatase (AP), Bilirubin, Lactate dehydrogenase (LDH) and total protein were measured by standard enzymatic method using a Vitros 250 analyzer (Ortho-Clinical-Diagnostics; Raritan, NJ, USA). 

### 3.8. Quantification and Statistical Analysis

We performed a One-way ANOVA analysis between and within group comparisons and corrected for multiple comparison by means of a Tukey Posthoc test. Outliers were identified and excluded based on the Robust regression and Outlier removal methods with Q1%. The resulting sample sizes after outlier exclusion is shown in the according dotplots. All data are indicated as means with standard errors of the mean except indicated otherwise. For analysis, GraphPad Prism 6.0 software; San Diego, CA, USA was used.

## 4. Discussion

Ischemia/reperfusion injury (IRI) occurs in several clinical settings, including hepatic surgery, liver transplantation, and haemorrhagic shock. Impaired liver function after I/R injury goes along with an inflammatory response that may cause significant cellular damage and organ dysfunction. Based on the current evidence about the organoprotective effects of argon in renal and neuronal cells, this noble gas appeared to be a promising candidate to reduce hepatic IRI [17,18,19,20,21,22].

To our knowledge, this is the first study investigating the effect of argon inhalation on hepatic IRI by measuring apoptosis, regenerative capacity, the modulation of the inflammatory response, and the effect on laboratory parameters. The main effect of IRI on the liver was noticed after 36 h in this study. At this time point, the Mitotic Index was significantly higher in the control group when compared to the argon group. In line with these findings, KI-67 was significantly higher in the control group, suggesting lower regenerative potential in the argon group. A similar effect was observed for BrdU, even though statistical significance was not reached. Conversely, there was a significantly higher expression of IL-1β and IL-6 after 36 h in the argon group compared to the control group. Both genes have been described to be crucial for the inflammatory response in liver injury. Together, these findings demonstrate that treatment with argon increases the pro-inflammatory response and reduces regenerative capacity after IRI. In line with this, aggravating effects of the noble gas argon on the inflammatory response have been reported earlier in the setting of myocardial ischemia/reperfusion [13].

The cytokines IL-1β, IL-6, and TNF-α are released by hepatic Kupffer cells and endothelial cells in liver injury [2,30,31]. This is of interest since IL-1β and TNF-α appear to be the protagonists in hepatic IRI [3,32,33,34,35]. Mechanisms of action include mediation of neutrophil adhesion and transmigration [2,3,36,37] and direct triggering of apoptosis [30]. Additionally, IL-1β enhances neutrophil-mediated production of free radicals [37,38]. In line with these findings, survival of IL-1β knockout mice has been shown to be superior to wild type mice after hepatic failure induced by administration of D-galactosamine together with endotoxin, suggesting an aggravating role of IL-1β in liver injury [39]. Consequently, administration of IL-1β-antagonists proved to be protective in a mouse model of acute alcoholic hepatitis [40]. 

The role of IL-6 in hepatic regeneration after injury appears to be unclear [41]. Several studies have assigned hepatoprotective effects to IL-6 after liver injury [42,43,44,45,46], while others report a hepatotoxic effect [47]. Most likely this contradiction is explained by the concentration of IL-6. While physiologic concentrations of IL-6 have hepatoprotective effects, hyperstimulation with IL-6 has been shown to impair liver regeneration [48]. 

Interestingly, we found the highest expression of IL-1β, IL-6 and TNF-α after 36 and 96 h, suggesting an impact of the type of liver injury on the different mechanism of regeneration. 

In line with the proinflammatory effect of argon claimed in this study, higher levels of Ki-67 were seen in the control group in comparison to the argon group, suggesting lower levels of liver regeneration after argon inhalation. There has been no difference in the number of apoptotic cells measured by TUNEL staining neither between different time points nor between the control and argon group in our study. This is of interest, as one would expect that clamping of the hepatic artery results in a significant amount of cell death in the obstructed liver lobe. One explanation could be that recent studies have suggested a leading role for oncotic necrosis instead of apoptosis in hepatic IRI with no significant change of TUNEL+ cells in longer periods of ischemia [30,49]. In line with this, we did not find a difference neither in AP, Bilirubin, LDH, protein or AST and ALT between argon group and control group. There were, however, significant elevations of the liver specific laboratory values AST and ALT when compared to sham-operated animals, indicating manifest hepatic IRI in our model.

There are several limitations to this study. First, there is still a lack of systematic studies that address a dose-dependent effect of argon in general and on liver IRI in particular. It thus remains speculative as to whether a better effect on IRI could be achieved using a different argon concentration and/or duration of application. Furthermore, in our study setting we tested the influence of argon after an ischemia-time of 45 min. A longer ischemic time might cause more pronounced effects. It would be desirable in the future to apply standardized protocols for the evaluation of the effects of argon in order to facilitate a better comparison of the data. Last, in extension to the measured AST and ALT levels, a more comprehensive evaluation of the functional outcomes such as short- and long-term survival after liver injury should be part of future studies. Furthermore, for a deeper understanding of the underlying mechanisms of hepatic regeneration after IRI, evaluation of protein levels such as HIF1-α might be an objective for further studies [27]. 

To summarize, liver IRI is a common and severe condition and therapeutic strategies are required. In this study, we could not find hepatoprotective effects of the noble gas argon, while we were able to show that on the contrary, argon seems to aggravate hepatic IRI via up-regulation of IL-1 β and IL-6. Consequently, the mitotic index was lower in the argon group than in the control group which was reflected in lower levels of Ki-67. Together, these findings suggest an impaired regeneration rather than the expected hepatoprotective effect of argon. 

## 5. Conclusions

This is the first study to describe the effects of argon on liver regeneration after warm I/R of the liver in rats. Contrary to a described cytoprotective effect of argon on other organs, argon aggravated the inflammatory response in the liver via up-regulation of IL-1β and IL-6 after 36 h. This was reflected in lower levels of Ki-67 and a lower mitotic index. These findings pave the way to further understanding of underlying mechanisms and therapeutic options in liver IRI. 

## Figures and Tables

**Figure 1 ijms-21-05457-f001:**
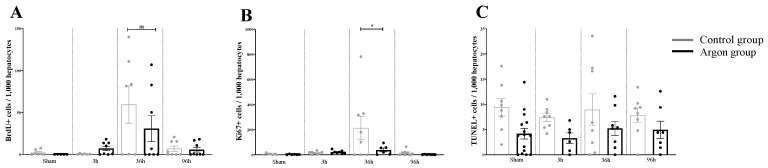
Quantitative analysis of immunohistochemistry data. (**A**) the number of BrdU positive cells/1000 hepatocytes is shown for all experimental conditions. A significant increase in the number of BrdU positive cells compared to all groups was observed in the control group after 36 h (significance bars are not indicated). Yet, this increase was not significantly different from the Argon group at 36 h (ns = not significant). (**B**) the number of Ki67+ cells/1000 hepatocytes is shown. The increase in Ki67+, which occurs in the control group at 36 h is prevented by Argon treatment, shown by a significant difference between the Argon and control group at 36 h (* = *p* < 0.05). (**C**) the number of TUNEL+ cells is shown. There were no significant differences between the experimental groups.

**Figure 2 ijms-21-05457-f002:**
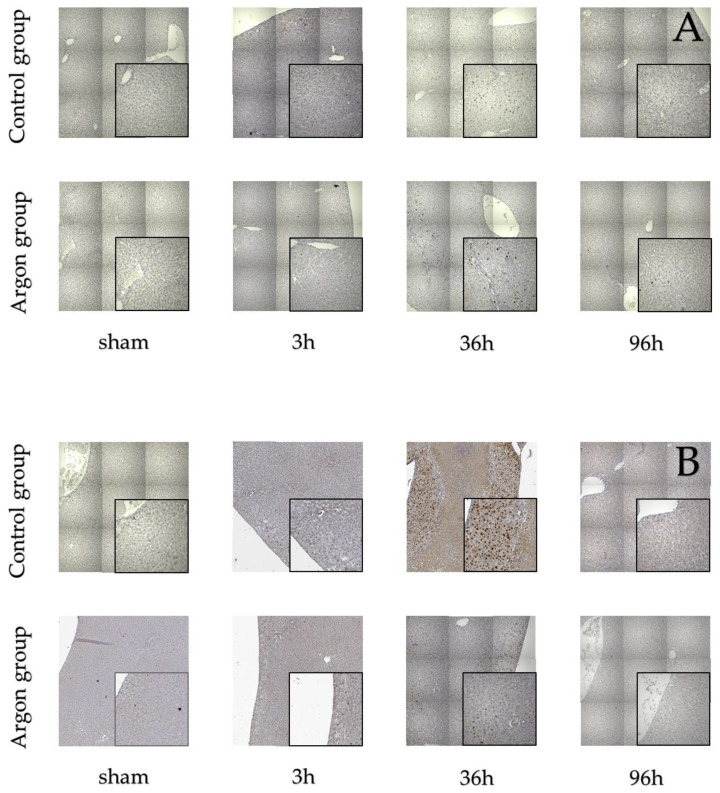
Immunostainings of BrdU (**A**), Ki-67 (**B**) and TUNEL (**C**).

**Figure 3 ijms-21-05457-f003:**
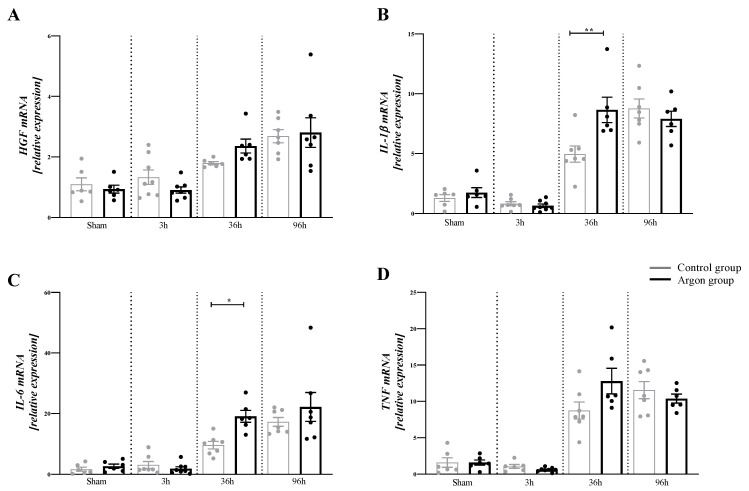
Relative mRNA expression of HGF (**A**), IL-1β (**B**), IL-6 (**C**), and TNF-α (**D**). There was a significant increase in the mRNA amount of IL-1β and IL-6 in the Argon treated group after 36 h. Only significant difference between the argon and control group are shown (* = *p* ≤ 0.05, ** = *p* < 0.001). Significant differences between time points are omitted.

**Figure 4 ijms-21-05457-f004:**
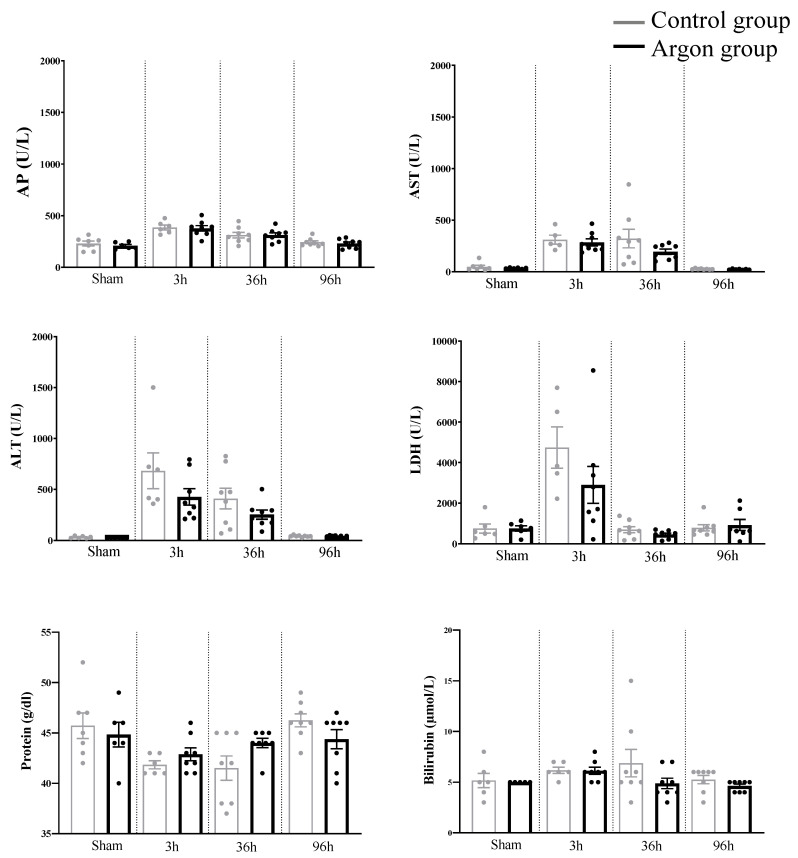
Laboratory parameters assessed in the different groups. There were no statistically significant differences between the argon and control group at any time point. Abbreviations: alkaline phosphatase (AP), aspartate aminotransferase (AST), alanine aminotransferase (ALT), Lactate dehydrogenase (LDH).

**Figure 5 ijms-21-05457-f005:**
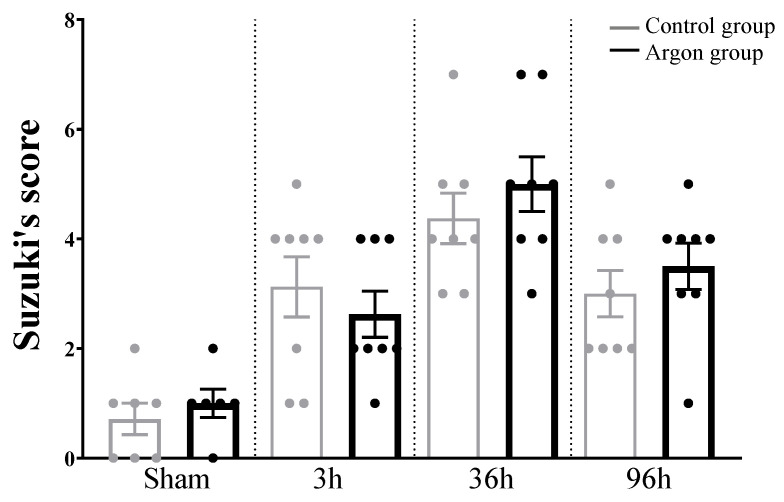
Liver damage evaluated by Suzuki’s score. There were no statistically significant differences between the argon and control group at any time point. Significant differences between time points are omitted.

**Figure 6 ijms-21-05457-f006:**
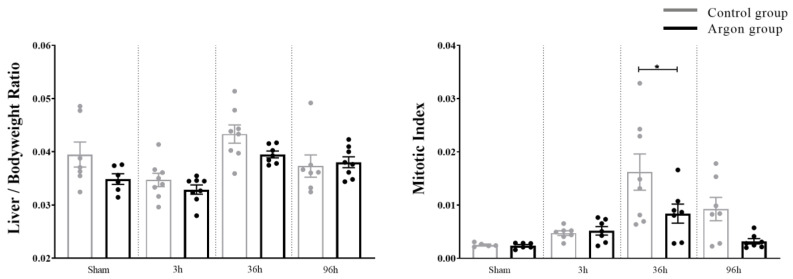
Liver regeneration evaluated by Liver/Body weight Ratio and Mitotic Index. Mitotic Index was significantly higher in the control group after 36 h (* = *p* < 0.05). Significant differences between time points are omitted.

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
