# Peer review of "Inhaled Argon Impedes Hepatic Regeneration after Ischemia/Reperfusion Injury in Rats"

_ijms, 2020, doi:10.3390/ijms21155457_

Round 1
Reviewer 1 Report
This is a manuscript describing the effect of argon on hepatic ischemia-reperfusion injury (IRI) in a rat model. Authors analyzed the effect of argon on IRI, however, there was no protective effect on hepatocytes in this study. Moreover, there was no definitive evidence that inhaled argon impedes hepatic regeneration after IRI, the title of this study. There are a lot of technical problems in this study.
Specific points.
#1. The main message and conclusion of this study were unclear. If the aim of this study is to determine the effect of argon on IRI, a lot of experiments should be added to address this issue as described below. If the aim of this study is to determine the effect of argon on hepatocyte proliferation after IRI, a lot of experiments should be added to address this issue. A previous report showed that argon delays initiation of liver regeneration after partial hepatectomy in rats. Authors should show the clear novelty in this study.
#2.To determine the effect of argon on IRI, authors should add the data related to hepatic IRI models as below: HE staining and scoring such as Suzuki’s score (ref.PMID: 29742804); infiltration of inflammatory cells using tissue staining or flowcytometry of immunocompetent cells; ELISA of serum cytokines and chemokines including IL-6, TNF-alpha, MCP1, and CXCL1; immunoblot of HNF4alpha, HIF1alpha, and NFkB-related molecules. It was unclear whether inhalation of argon change hepatocyte viability, metabolism, and inflammation in liver after IRI in this study.
#3. The conclusion of this study was that inhalation of argon inhibited hepatocyte proliferation after IRI, however, the suppression of hepatocyte proliferation by argon inhalation was very unclear. There was no significant difference in BrdU incorporation between control and argon group (Figure 1A). Authors should precisely analyze this point and add the definitive data about proliferation and apoptosis of hepatocytes, including the data about PCNA, cyclin family, and caspase-3 using immunoblot, and related data about serum IL-6, TNF-alpha, MCP1, and CXCL1 using ELISA. Expression profile of both groups should be assessed by comprehensive analysis such as microarray. Authors should examine the molecular mechanism that inhalation of argon inhibits hepatocyte proliferation. Gain of function assay will be needed, which shows that a inhibitor of a specific pathway cancels the effect of argon inhalation.
#4. The quality of photograph in Figure 2 should be improved. Inset of high-power-field should be added.
#5. Figure 4. Several biochemical tests showed the tendency that argon inhalation may reduce serum ALT (3h and 36h) and serum LDH (3h). Did the authors check other time-point?
Reviewer 2 Report
In this manuscript, Schmitz et al. explored the impact of argon on liver regeneration after ischemia/reperfusion injury (IRI). The argon inhalation decreased the percentage of Ki-67 positive cells after IRI. Proinflammatory cytokines IL-1β and IL-6 were also elevated after argon inhalation. However, apoptosis and liver function test showed no differences between groups.
- In figure 1A, the percentage of TUNEL positive cells are similar in sham group and IRI. Why does IRI not increase the number of TUNEL positive cells? If apoptosis is not the main mechanism of cell death during hepatic IRI, please showed the necrosis results.
- As provided from the data, the argon groups did not show worser liver function or more cell death. Therefore, we can not conclude that argon inhalation is detrimental on liver regeneration after IRI.
- Whether or not the argon inhalation group had more injury is important and needs to be clarify. More proliferation (BrdU or Ki-67+) could stand for better regeneration for sure. On the other hand, it can also suggest less injury happened. The control and argon groups could have different responses to the injury. Different levels of injury may result in different levels of proliferation demands.
- How about long-term fibrogenesis between groups after IRI?
- In figure 4, the results of ALT (3h and 96h) were either labeling wrong or flipped.
Round 2
Reviewer 1 Report
This is a revised manuscript describing the effect of argon on hepatic ischemia-reperfusion injury (IRI) in a rat model. Authors added several data and revised several points, however, several analysis are needed to prove that argon inhalation impedes hepatic regeneration after IRI.
Major points.
#1. To determine the effect of argon on IRI, authors should add the data of ELISA of cytokines and chemokines in serum and/or livers, including IL-6, TNF-alpha, MCP1, and CXCL1. It is because authors concluded that argon inhalation has detrimental effects on liver regeneration after IRI related to elevated levels of IL-1beta and IL-6. Protein level assay should be added (same as Previous Major point #2).
#2. To assess whether inflammation and regeneration are changed by argon inhalation, authors should add the protein level data about HNF4alpha, HIF1alpha, and NFkB-related molecules (same as Previous Major point #2).
#3. The suppression of hepatocyte proliferation by argon inhalation was very unclear(same as Previous Major point #3). To determine the effect of argon on hepatocyte proliferation after IRI, Authors should add the data about proliferation and apoptosis of hepatocytes, including the data about PCNA, cyclin family, and Caspase-3 using protein level assay.
Minor points.
#4. Figure2. Inset of high-power-field should be added. It is not appropriate that the contrasts of photographs are changed electronically (same as Previous point #4).
Author Response
This is a revised manuscript describing the effect of argon on hepatic ischemia-reperfusion injury (IRI) in a rat model. Authors added several data and revised several points, however, several analysis are needed to prove that argon inhalation impedes hepatic regeneration after IRI.
Major points.
#1. To determine the effect of argon on IRI, authors should add the data of ELISA of cytokines and chemokines in serum and/or livers, including IL-6, TNF-alpha, MCP1, and CXCL1. It is because authors concluded that argon inhalation has detrimental effects on liver regeneration after IRI related to elevated levels of IL-1beta and IL-6. Protein level assay should be added (same as Previous Major point #2).
#2. To assess whether inflammation and regeneration are changed by argon inhalation, authors should add the protein level data about HNF4alpha, HIF1alpha, and NFkB-related molecules (same as Previous Major point #2).
#3. The suppression of hepatocyte proliferation by argon inhalation was very unclear(same as Previous Major point #3). To determine the effect of argon on hepatocyte proliferation after IRI, Authors should add the data about proliferation and apoptosis of hepatocytes, including the data about PCNA, cyclin family, and Caspase-3 using protein level assay.
Minor points.
4. Figure2. Inset of high-power-field should be added. It is not appropriate that the contrasts of photographs are changed electronically (same as Previous point #4).
Dear Reviewer,
thank you very much again for your helpful comments!
For this project, we chose PCR analysis for IL-6, IL-1beta, HGF and TNF. It is true that based on this analysis, the conclusion was drawn mainly on the difference between IL-6 and IL-1beta. However, the same trend could be observed for TNF, suggesting an overall higher inflammation in argon inhalation. Furthermore, at the same time point (36 hours), Ki-67 was increased in the control group and a trend towards an increased BrdU expression could seen in the control group. We added the liver / body weight ratio and the mitotic index to the manuscript and the findings of a suppressed liver regeneration in argon inhalation are reflected in an impaired mitotic index. Adding the other molecules suggested would result in an entirely new project. However, we believe that our findings are congruent with the clinical result (mitotic index).
The fact that assessment of HNF4alpha, HIF1alpha, and NFkB-related molecules might faciliate understanding of underlying mechanisms of argon inhalation was added to the manuscript as a limitation point in the discussion. We believe there is room for further studies here. Again, we appreciate the work done on this field before but we cannot provide these investigations at this time point.
We have to apologize for the changes in the histological stainings - that was a misunderstanding. We now recaptured all of the stainings and added HPF insets.
Again, thank you very much for your valuable comments.
Reviewer 2 Report
The authors fully addressed my questions.
Author Response
Thank you very much for your appreciation of our manuscript!
Round 3
Reviewer 1 Report
Authors have revised several points.